# First Virtual Reconstruction of a Mosasaurid Brain Endocast: Description and Comparison of the Endocast of *Tethysaurus nopcsai* with Those of Extant Squamates

Rémi Allemand [1,*], Michael J. Polcyn [2,3], Alexandra Houssaye [4], Peggy Vincent [1], Camilo López-Aguirre [5] and Nathalie Bardet [1]

1   CR2P Centre de Recherche en Paléontologie de Paris, UMR 7207 CNRS-MNHN-Sorbonne Université, Muséum National d'Histoire Naturelle, 57 Rue Cuvier CP-38, 75005 Paris, France; peggy.vincent@mnhn.fr (P.V.); nathalie.bardet@mnhn.fr (N.B.)

2   Faculty of Geosciences, Utrecht University, Princetonlaan 8a, 3584 CB Utrecht, The Netherlands; m.j.polcyn@uu.nl

3   Huffington Department of Earth Sciences, Southern Methodist University, 3225 Daniel Ave., Dallas, TX 75275, USA; mpolcyn@smu.edu

4   MECADEV Mécanismes Adaptatifs et Évolution, UMR 7179 CNRS-MNHN-Sorbonne Université, Muséum National d'Histoire Naturelle, 57 Rue Cuvier CP-55, 75005 Paris, France; alexandra.houssaye@mnhn.fr

5   Department of Anthropology, University of Toronto Scarborough, 1265 Military Trail, Toronto, ON M1C 1A4, Canada; c.lopezaguirre@utoronto.ca

*   Correspondence: remi.allemand1@mnhn.fr

**Abstract:** Paleoneurological studies of mosasaurids are few and limited to old partial reconstructions made from latex casts on *Platecarpus* and *Clidastes*. Here, the brain endocasts of three specimens of the early mosasaurid *Tethysaurus nopcsai* from the Turonian of Morocco are reconstructed for the first time by using micro-computed tomography. Comparisons between *Tethysaurus* and the later *Platecarpus* and *Clidastes* show that distinct endocranial organizations have occurred within the clade through time, including differences in the flexure of the endocast and the size of the parietal eye. The physiological consequences of such variability remain unclear and further investigations are required to better interpret these variations. In addition, the endocast of *Tethysaurus* was compared to those of extant anguimorphs, iguanians, and snakes, using landmark-based geometric morphometrics. The results revealed that *Tethysaurus* exhibits a unique combination of endocranial features compared to extant toxicoferans. Contrary to previous statements, we find no strong resemblance in endocast morphology between *Tethysaurus* and varanids. Rather, the endocast of *Tethysaurus* shows some morphological similarities with each of the clades of anguimorphs, iguanians, and snakes. In this context, while a notable phylogenetic signal is observed in the variability of squamate endocasts, it is premature to establish any phylogenetic affinities between mosasaurids and extant squamates based solely on endocast morphologies.

**Keywords:** mosasaurids; squamates; brain endocast; landmarks; geometric morphometrics

## 1. Introduction

Mosasauridae is a clade of squamates that became secondarily adapted for marine life in the early Late Cretaceous and went extinct at the end of the Maastrichtian, during the K/Pg biological event [1,2]. During their existence, mosasaurids showed increasing adaptations to the marine environment through time [1]. They became increasingly efficient swimmers capable of deep prolonged repetitive diving (e.g., [3,4]) and thrived in many marine habitats from rocky shores to open oceans, including freshwater environments [2]. Mosasaurids were very diversified and occupied a wide range of ecological niches, showing a wide range of body sizes, locomotor styles, diets, and feeding strategies (e.g., [5–11]). By the end of the Creta-ceous, they were the apex predators in many marine ecosystems and attained a cosmopolitan

distribution (e.g., [1,2,12–14]). Fossil remains of mosasaurids have been recovered from all continents and from latitudes near the Arctic [15] to the Antarctic [16–18]. Mosasauridae comprises at least four subfamilies: Halisaurinae, Mosasaurinae, Tylosaurinae, and Plioplatecarpinae [19], although the details of mosasaurid relationships may vary depending on authors and phylogenetic analyses (e.g., [20–22]). Mosasaurids include "early diverging" taxa that exhibit plesiomorphic morphological characteristics (i.e., *Tethysaurus*, *Russellosaurus*, *Dallasaurus*, *Yaguarasaurus*, and Halisaurinae) and "later diverging" species (i.e., remaining members of the subfamilies Mosasaurinae, Plioplatecarpinae, and Tylosaurinae), which are morphologically more derived (see [19,23] for more details).

Recently, several paleoneurological studies have used non-invasive imaging techniques to explore mosasaurid internal cranial structures, such as the endosseous labyrinth [24–26] and the neurovascular system [23,27]. However, such paleoneurological investigations remain rare and there are currently no studies delving into the mosasaurid neuroanatomy through computed tomographic scanning. Indeed, brain endocasts in mosasaurids are only known from two latex casts of the endocranial cavity performed by Camp [28] for *Clidastes propython* (UCMP 34535; see Figure 1a) and *Platecarpus* sp. (UCMP 34781; see Figure 1b). The partial brain endocast in *Platecarpus* was reconstructed by joining the dissociated braincase elements and by filling the brain cavity "with liquid rubber backed with sawdust" [28] (p. 40). Due to the open condition of the endocranial cavity, the antero-ventral extension of the *Platecarpus* endocast could not be reconstructed; however, Camp's work did provide valuable information regarding its global morphology [28]. In *Clidastes*, the endocast was reconstructed by using a latex mold on the ventral surfaces of the frontal and parietal only [28] (p. 40). As a result, only the antero-dorsal part of the endocast was reconstructed and Camp provided no additional details about the rest of the structure. Based on these reconstructions, Camp [28] conducted a comparison between the endocasts of *Platecarpus* and *Clidastes* with the brain of a juvenile *Varanus niloticus*. As he observed a significant resemblance among the three species, Camp [28] suggested a close phylogenetic relationship between mosasaurids and varanids.

Several studies have shown that both brain and endocast morphologies in squamates reflect a phylogenetic signal (e.g., [29–33]). However, it is worth noting that Camp's comparison, by mixing brain and endocast, might have introduced some biases. Indeed, it is now known that the brain in squamates fits into the endocranial cavity in ways that vary depending on taxa and ontogeny (e.g., [34–39]). Thus, brains and endocasts in squamates should be considered as distinct structures and not directly compared in order to avoid misinterpretations [40]. In this context, assessing the validity of Camp's hypothesis would require comparing the mosasaurid endocast with that of an adult *Varanus*. In addition, given the lack of consensus regarding the phylogenetic relationships of mosasaurids within Squamata (e.g., [19,41–47]), expanding the scope of comparison to include endocasts of other extant squamates would be more suitable for assessing endocranial morphological affinities between mosasaurids and other squamates.

Using micro-computed tomography, the brain endocast of the early mosasaurid *Tethysaurus nopcsai* [48] from the Turonian of the Goulmima area, southern Morocco, is reconstructed here, described, and compared to the latex reconstitutions made by Camp [28] for the later diverging mosasaurids *Platecarpus* and *Clidastes*. In addition, the endocast of *Tethysaurus* is compared to those of extant squamates of the clade Toxicofera, including anguimorphs, iguanians, and snakes (e.g., [44]). Using landmark-based geometric morphometrics, this study aims to determine which extant squamates the endocast of *Tethysaurus* most closely resembles. The results obtained will allow us to assess the validity of Camp's hypothesis and to discuss the phylogenetic and biological implications.

Institutional Abbreviations—CAS, California Academy of Science, San Francisco, California, USA; FMNH, The Field Museum of Natural History, Chicago, Illinois, USA; KU, University of Kansas, Lawrence, Kansas, USA; LSUMZ, Louisiana State University Museum of Natural Science, Bâton Rouge, Louisiana, USA; MNHN, Muséum National d'Histoire Naturelle, Paris, France; MVZ, Museum of Vertebrate Zoology, University of California, Berkeley, USA; NCSM, North Carolina Museum of Natural Sciences, Raleigh, North

Carolina, USA; TCWC, Texas Cooperative Wildlife Collection, Department of Wildlife Science, Texas A&M University, College Station, Texas, USA; TMM, Texas Memorial Museum, Austin, Texas, USA; TNHC, Texas Natural History Collections, Austin, Texas, USA; SMU, Shuler Museum of Paleontology, Southern Methodist University, Dallas, Texas, USA; UCMP, University of California Museum of Paleontology, Berkeley, CA, USA; UF, The University of Florida, Gainesville, Florida, USA; UMMZ, University of Michigan Museum of Zoology, Ann Arbor, Michigan, USA; UTA, The University of Texas at Arlington, Arlington, Texas, USA; YPM, Yale Peabody Museum, New Haven, Connecticut, USA; ZRC, Zoological Reference Collections, National University of Singapore, Kent Ridge, Singapore.

## 2. Materials and Methods

### 2.1. Specimen Sampling and Data Acquisition

The three specimens of the mosasaurid *Tethysaurus nopcsai* analyzed here come from the Turonian in age Unit T2a of the Akrabou Formation, in Goulmima area, Er-Rachidia Province, in southern Morocco [48].

The holotype specimen MNHN GOU 1 was described by Bardet et al. [48] and consists of a nearly complete articulated skull and mandible. Although it is preserved in three dimensions (see Figure 1 in [48]), it is slightly crushed laterally, resulting in the displacement of some bones from their original position. The specimen was scanned at the AST-RX platform of the MNHN (Paris, France) using a v | tome | x L240-180 computed tomograph model from Baker Hughes Waygate Technologies (Huerth, Germany) and reconstructions were performed using DATOX/RES software (phoenix datos | x).

The two other specimens, SMU 76335, which is unpublished, and SMU 75486, of which aspects of the snout and circumorbital series are published [19], consist of nearly complete articulated skulls and mandibles preserved in 3D. They were scanned at the University of Texas High-Resolution X-ray CT Facility (Austin, TX, USA) using an NSI scanner and a 210-kV Feinfocus microfocal source. Voxel size naturally varies depending on specimen size (see Table 1).

Seventy-seven extant species of the clade Toxicofera (See Table 1 and Figures S1–S3) were chosen for the purpose of conducting comparisons with the endocast of *Tethysaurus*. The sample includes 13 iguanians (Figure S1), 29 anguimorphs (Figure S2), and 35 snakes (Figure S3) that were selected to grossly represent the taxonomic diversity of these groups. Computed tomographic scans of these species were obtained from different sources. Scans for eight specimens were sampled from the previous work of Allemand et al. [29] and 69 specimens were acquired from the online database MorphoSource [49] "http://www.MorphoSource.org/" (accessed on 3 August 2024).

Image segmentation and visualization were performed using the software Avizo version 2019.1 (Thermo Fisher Scientific, Waltham, MA, USA). The segmentation tools were used to manually reconstruct the endocast for each species by segmenting the internal surface of the bones or the dura mater when no bones surround the endocranial cavity.

**Table 1.** Toxicofera list of species analyzed. See Institutional Abbreviations for collection numbers. AH-unnumb, Anthony Herrel (UMR 7179, CNRS/MNHN, Paris, France) personal collection. Ab., Species name abbreviations used in Figure 2. An asterisk (*) indicates specimens coming from the MorphoSource database. Taxonomic classification from [50–52].

| Suborder | Family | Species | Ab. | Collection Number | Voxel Size (mm) |
|---|---|---|---|---|---|
| | | *Abronia deppii* | Ab.d | CAS:herp:143109 * | 0.037 |
| | | *Abronia graminea* | Ab.g | UTA:uta-r:38831 * | 0.014 |
| | | *Abronia taeniata* | Ab.t | TCWC:herpetology:4911 * | 0.015 |
| | | *Anguis fragilis* | An.f | MVZ:herp:238523 * | 0.044 |
| | | *Barisia imbricata* | Ba.i | TNHC:herpetology:76984 * | 0.014 |
| Anguimorpha | Anguidae | *Dopasia harti* | Do.h | NCSM:herp:80838 * | 0.042 |
| | | *Elgaria kingii* | El.k | UF:herp:74645 * | 0.033 |
| | | *Gerrhonotus infernalis* | Ge.i | TNHC: herpetology:92262 * | 0.026 |
| | | *Mesaspis moreletii* | Me.m | UF:herp:51455 * | 0.022 |
| | | *Ophisaurus mimicus* | Op.m | NCSM:herp:25699 * | 0.021 |
| | | *Pseudopus apodus* | Ps.a | KU:kuh:87837 * | 0.07 |

**Table 1.** *Cont.*

| Suborder | Family | Species | Ab. | Collection Number | Voxel Size (mm) |
|---|---|---|---|---|---|
| Anguimorpha | Anniellidae | *Anniella grinnelli* | An.g | MVZ:herp:257738 * | 0.021 |
| | Diploglossidae | *Celestus costatus* | Ce.c | UF:herp:59382 * | 0.027 |
| | | *Celestus hylaius* | Ce.h | UF:herp:75794 * | 0.038 |
| | | *Diploglossus fasciatus* | Di.f | UMMZ:herps:115647 * | 0.058 |
| | | *Ophiodes striatus* | Op.s | YPM:vz:ypm herr 013348.001 * | 0.036 |
| | Xenosauridae | *Xenosaurus grandis* | Xe.g | FMNH:Amphibians and Reptiles:123702 * | X, Y = 0.027/Z = 0.064 |
| | | *Xenosaurus platyceps* | Xe.p | UTA:uta-r:23594 * | X, Y = 0.023/Z = 0.053 |
| | Helodermatidae | *Heloderma horridum* | He.h | UF:herp:42033 * | 0.047 |
| | Varanidae | *Varanus acanthurus* | Va.a | UTA:uta-r:13015 * | X, Y = 0.023/Z = 0.051 |
| | | *Varanus exanthematicus* | Va.e | AH_unnumb | 0.045 |
| | | *Varanus gouldii* | Va.g | TMM:m:1295 * | X, Y = 0.084/Z = 0.21 |
| | | *Varanus komodoensis* | Va.k | TNHC:herpetology:95803 * | 0.163 |
| | | *Varanus niloticus* | Va.n | UF:herp:83764 * | 0.041 |
| | | *Varanus prasinus* | Va.p | UF:herp:71411 * | 0.037 |
| | | *Varanus salvator* | Va.s | FMNH:Amphibians and Reptiles:35144 * | X, Y = 0.088/Z = 0.201 |
| | | *Varanus timorensis* | Va.t | UF:herp:137865 * | 0.058 |
| | Lanthanotidae | *Lanthanotus borneensis* | La.b | FMNH:Amphibians and Reptiles:148589 * | X, Y = 0.022/Z = 0.046 |
| | Shinisauridae | *Shinisaurus crocodilurus* | Sh.c | FMNH:Amphibians and Reptiles:215541 * | X, Y = 0.029/Z = 0.078 |
| Serpentes | Anomalepididae | *Typhlophis squamosus* | Ty.s | MNHN 1997.2042 | 0.005 |
| | Typhlopidae | *Acutotyphlops kunuaensis* | Ac.k | LSUMZ:herps:93566 * | 0.019 |
| | | *Amerotyphlops brongersmianus* | Am.b | FMNH:Amphibians and Reptiles:195928 * | 0.033 |
| | | *Typhlops arenarius* | Ty.a | UMMZ:herps:241854 * | 0.01 |
| | Aniliidae | *Anilius scytale* | An.s | MNHN 1997.2106 | 0.01 |
| | Tropidophiidae | *Tropidophis canus* | Tr.c | UMMZ:herps:117024 * | 0.017 |
| | Boidae | *Boa constrictor* | Bo.c | FMNH:Amphibians and Reptiles:31182 * | X, Y = 0.078/Z = 0.174 |
| | | *Candoia carinata* | Ca.c | LSUMZ:herps:93576 * | 0.035 |
| | | *Eunectes murinus* | Eu.m | UF:herp:84822 * | 0.074 |
| | | *Sanzinia madagascariensis* | Sa.m | KU:kuh:183837 * | 0.055 |
| | Cylindrophiidae | *Cylindrophis ruffus* | Cy.r | UF:herp:143722 * | 0.040 |
| | Uropeltidae | *Rhinophis sanguineus* | Rh.s | UF:herp:78397 * | 0.022 |
| | Pythonidae | *Morelia spilota* | Mo.s | UMMZ:herps:227833 * | 0.054 |
| | | *Python bivittatus* | Py.b | UF:herp:167549 * | 0.086 |
| | | *Python molurus* | Py.m | UF:herp:190353 * | 0.052 |
| | Acrochordidae | *Acrochordus javanicus* | Ac.j | KU:kuh:318186 * | 0.025 |
| | Viperidae | *Bitis arietans* | Bi.a | UMMZ:herps:61258 * | 0.021 |
| | | *Crotalus molossus* | Cr.m | UMMZ:herps:143742 * | 0.017 |
| | | *Vipera aspis* | Vi.a | UMMZ:herps:116957 * | 0.019 |
| | Homalopsidae | *Cerberus rynchops* | Ce.r | MNHN-RA-1998.8583 | 0.035 |
| | | *Gerarda prevostiana* | Ge.p | CAS:herp:204972 * | 0.015 |
| | | *Homalopsis buccata* | Ho.b | ZRC 2.6411 | 0.024 |
| | Atractaspididae | *Atractaspis bibronii* | At.b | UMMZ:herps:209986 * | 0.012 |
| | Elapidae | *Aipysurus duboisii* | Ai.d | MNHN-RA-1990.4519 | 0.041 |
| | | *Bungarus fasciatus* | Bu.f | UMMZ:herps:201916 * | 0.019 |
| | | *Emydocephalus annulatus* | Em.a | UMMZ:herps:93851 * | 0.022 |
| | | *Hydrophis platurus* | Hy.p | AH_MS 64 | 0.032 |
| | | *Hydrophis schistosus* | Hy.s | ZRC 2.2043 | 0.021 |
| | | *Laticauda colubrina* | La.c | UMMZ:herps:65950 * | 0.017 |
| | | *Naja nigricollis* | Na.n | UMMZ:herps:203814 * | 0.025 |
| | | *Pseudechis porphyriacus* | Ps.p | UMMZ:herps:170403 * | 0.026 |
| | Colubridae | *Afronatrix anoscopus* | Af.a | CAS:herp:230205 * | 0.015 |
| | | *Drymarchon corais* | Dr.c | UMMZ:herps:190326 * | 0.018 |
| | | *Lycodon striatus* | Ly.s | UMMZ:herps:123427 * | 0.012 |
| | | *Tropidonophis picturatus* | Tr.p | LSUMZ:herps:96093 * | 0.028 |
| Iguania | Agamidae | *Agama agama* | Ag.a | UF:herp:180711 * | 0.02 |
| | | *Draco volans* | Dr.v | UF:herp:48909 * | 0.018 |
| | | *Physignathus cocincinus* | Ph.c | YPM:vz:ypm herr 014378 * | X, Y = 0.023/Z = 0.055 |
| | Chamaeleonidae | *Chamaeleo calyptratus* | Ch.c | UF:herp:191369 * | 0.041 |
| | Iguanidae | *Amblyrhynchus cristatus* | Am.c | UF:herp:41558 * | 0.052 |
| | | *Ctenosaura similis* | Ct.s | UF:herp:181929 * | 0.061 |

**Table 1.** *Cont.*

| Suborder | Family | Species | Ab. | Collection Number | Voxel Size (mm) |
|---|---|---|---|---|---|
| Iguania | Phrynosomatidae | *Sceloporus undulatus* | Sc.u | NCSM:herp:83600 * | 0.016 |
| | | *Uta stansburiana* | Ut.s | FMNH:Amphibians and Reptiles:213914 * | X, Y = 0.014/Z = 0.036 |
| | Dactyloidae | *Anolis carolinensis* | An.c | UF:herp:102367 * | 0.013 |
| | Corytophanidae | *Basiliscus basiliscus* | Ba.b | FMNH:Amphibians and Reptiles:68188 * | 0.068 |
| | Hoplocercidae | *Enyalioides heterolepis* | En.h | UF:herp:68015 * | 0.021 |
| | Leiocephalidae | *Leiocephalus carinatus* | Le.c | UF:herp:185239 * | 0.029 |
| | Tropiduridae | *Stenocercus roseiventris* | St.r | KU:kuh:214966 * | 0.09 |
| Mosasauria | Mosasauridae | *Tethysaurus nopcsai* | Te.n | MNHN GOU 1<br>SMU 76335<br>SMU 75486 | 0.0814<br>0.0778<br>0.081 |

### 2.2. Landmarks and Statistical Analysis

To compare endocast morphologies, we employed the landmark protocol defined by Allemand et al. [30]. Of the twenty landmarks available in [30], nineteen were selected here. Landmark 8, defined as the 'most ventro-median extent of the endocast at the posterior margin of the optic nerve foramen' [30], could not be located in *Tethysaurus* due to the non-preservation of the orbitosphenoid bone. All landmarks were placed on the virtual endocasts and exported using the software Avizo (version 2019.1). Landmarks were only placed on one *Tethysaurus* specimen (SMU 76335) as the endocast is the most complete and is less deformed compared to the two other specimens (see Supplementary Data S1 for landmark description and position on *Tethysaurus* endocast).

We performed a Generalized Procrustes Analyses (GPA) by using the gpagen function in the R package 'geomorph' [53] to quantify and visualize differences in endocast morphologies captured by the landmarks (see Supplementary Data S2 for the raw coordinates). Phylogenetic structuring of the endocast morphology of extant toxicoferans was assessed by estimating the multivariate K-statistic using the physignal function in the 'geomorph' package [54]. The typology of the phylogenetic tree used to run these analyses is modified from [50,51]. Then, to estimate the occurrences of allometry, the relationships of endocast shape with size were tested based on log-transformed centroid size, using a Procrustes regression with the procD.lm function in 'geomorph' [55]. Given the strong statistical significance of allometry in endocast morphology, the residuals of the regression were used as allometry-corrected shape data in subsequent analyses. A Procrustes analysis of variance (PLM) using the procD.lm function of the 'geomorph' package was performed on allometry-corrected shape data to test patterns of endocast shape variation between iguanians, snakes, and anguimorphs. Pairwise comparisons in the allometry-corrected shape variance between extant toxicoferans were conducted using the pairwise function in the R package 'RRPP' [55]. All tests of statistical significance were based on the distribution of 10,000 iterations.

A Principal Component Analysis (PCA) using the gm.prcomp function in 'geomorph' was performed on the allometry-corrected shape data in order to visualize the pattern of endocranial shape variation in *Tethysaurus*, iguanians, anguimorphs, and snakes. A Linear Discriminant Analysis (LDA) was performed on the PC scores to highlight similarities in endocast morphology between *Tethysaurus* and the three clades of extant toxicoferans. The LDA was carried out using the lda function in the R package 'MASS' [56] on the ten first PCs (accounting for 92% of the variance, see Supplementary Data S3) in order to keep fewer variables than specimens from each group (13 iguanians, 29 anguimorphs, and 35 snakes). The accuracy of the LDA was tested using our dataset for extant species, classifying every species as either iguanian, anguimorph, or snake. Posterior probabilities of the LDA were used to determine to which clade the *Tethysaurus* endocast shows the closest resemblance among snakes, anguimorphs, and iguanians.

## 3. Results

### 3.1. Brain Endocast of Tethysaurus nopcsai

Among the three specimens of *Tethysaurus* used here, the brain endocast reconstructed in SMU 76335 is the most complete with almost all endocranial regions visible (Figure 1c,d). In the specimens SMU 75486 (Figure 1e,f) and MNHN GOU 1 (Figure 1g,h), only the olfactory bulbs and peduncles as well as the posterior part of the endocast could be reconstructed. Indeed, in both specimens, the state of preservation of the parietal bone prevents the lateral delimitation of the endocast in this area. The endocast in SMU 75486 (Figure 1e,f) is slightly crushed laterally, the antero-dorsal part of the endocast being not aligned with the posterior one. We refer to endocranial regions by their underlying soft-tissue features (i.e., "cerebral hemispheres" rather than "impression of the cerebral hemispheres").

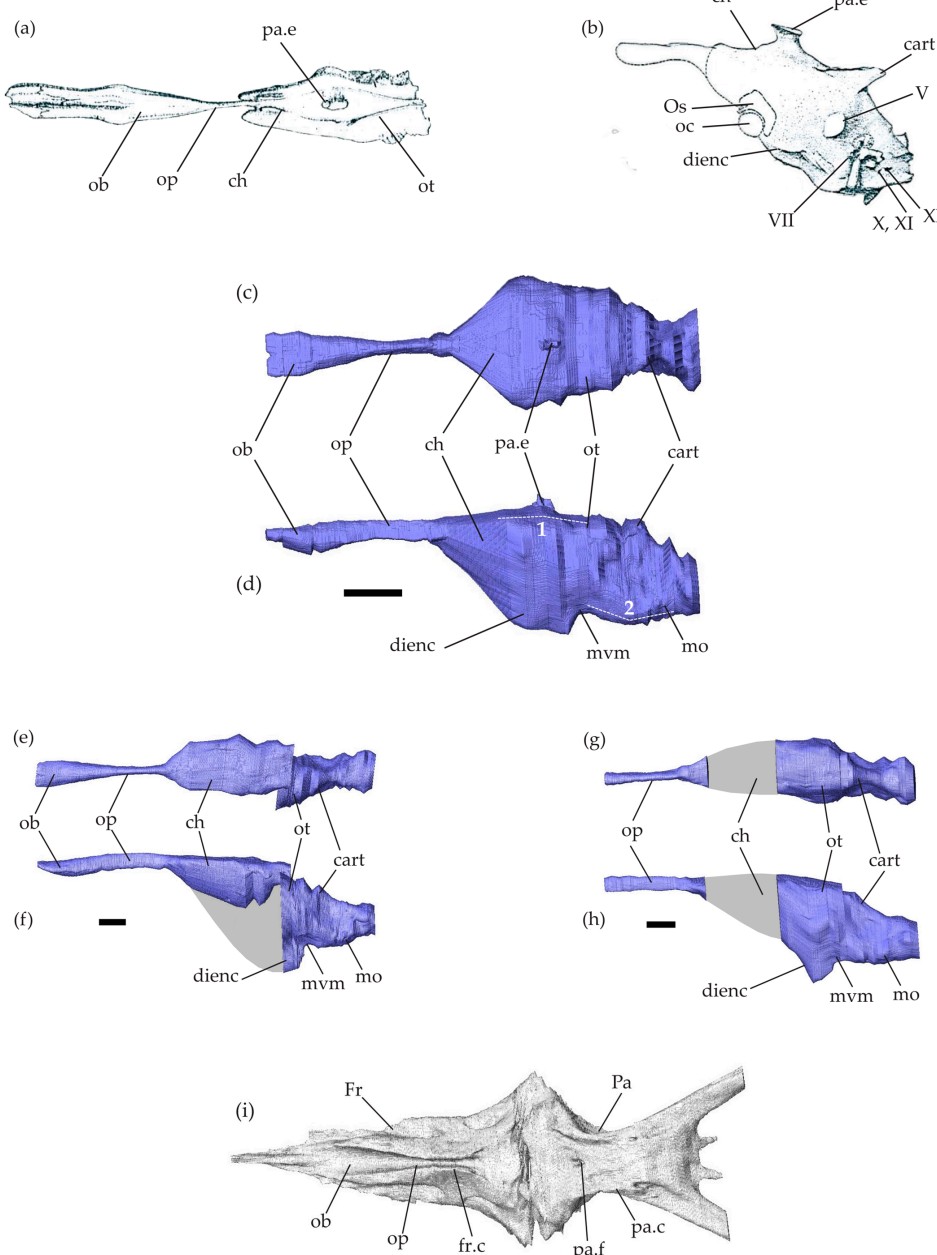

**Figure 1.** Mosasaurid brain endocasts. Brain endocasts of *Clidastes propython* UCMP 34535 in dorsal view (**a**) and *Platecarpus* sp. UCMP 34781 in left lateral view (**b**), modified from Camp [28] (no scale bar available). (**c–h**) Brain endocasts of *Tethysaurus nopcsai* specimen SMU 76335 (**c,d**), SMU 75486 (**e,f**),

and MNHN GOU 1 (**g**,**h**) in dorsal (**c**,**e**,**g**) and left lateral (**d**,**f**,**h**) views. Scale bars equal 10 mm. (**i**) Virtual reconstructions of the frontal and parietal bones in SMU 76335 in ventral view. Grey areas on the endocasts represent regions that could not be reconstructed. Abbreviations: 1, cephalic flexure; 2, pontine flexure; cart., cartilaginous bridge; ch, cerebral hemispheres; dienc, ventral diencephalon; Fr, frontal; fr.c, frontal cranial crests; mo, medulla oblongata; mvm, mesencephalic ventral margin; ob, olfactory bulbs; oc, optic chiasm; op, olfactory peduncles; Os, orbitosphenoid; ot, optic tectum; Pa, parietal; pa.c, parietal cranial crests; pa.e, parietal eye; pa.f, parietal foramen; V, trigeminal nerve; VII, facial nerve; X,XI, vagus and accessory nerves; XII, hypoglossal nerves.

The endocranial cavity in *Tethysaurus* is surrounded by several bones including the frontal and the parietal from the skull roof (Figure 1i) and the elements forming the braincase: para-basisphenoid, basioccipital, prootic, opisthotic–exoccipital, and supraoccipital. These bones delimit only the dorsal surface as well as the postero-lateral part of the endocranial cavity. The antero-ventral portion of the cavity remains non-ossified, preventing the accurate ventral delimitation of the cerebral hemisphere margins.

The *Tethysaurus* endocast follows the general organizations observed in squamates and other non-avian reptiles (e.g., [40,57]) in that it exhibits a tubular shape that is longer than wide and has a smooth surface (Figure 1b,c–h). In the lateral view, the endocast in *Tethysaurus* appears relatively narrow and flattened (Figure 1d,f,h), showing only weak cephalic and pontine flexures (i.e., angles formed between the telencephalon–mesencephalon and mesencephalon–rhombencephalon, respectively, [34]) that differ from the pronounced flexures figured in *Platecarpus* by Camp [28].

The anterior part of the endocast consists of the olfactory bulbs and the olfactory peduncles (Figure 1c–f). The external morphology of the endocast provides no information that allows the precise delimitation of the olfactory bulbs from the peduncles. The medial limit separating the paired olfactory bulbs and peduncles is not visible on the dorsal surface of the endocast. The olfactory bulbs and peduncles are elongated, representing nearly half of the total endocast length. This is similar to the proportions figured in *Clidastes* by Camp [28] (see Figure 1a), whereas in *Platecarpus* (Figure 1b), the structure seems to be relatively shorter [28,58]. As in *Clidastes* [28], the whole structure (i.e., olfactory bulbs and peduncles) in *Tethysaurus* is wider anteriorly than posteriorly, being mediolaterally compressed in the middle of the antero-posterior length (Figure 1c,e). The olfactory bulbs and peduncles in *Tethysaurus* are roofed by the frontal bone and correspond to the fossa visible on the ventral surface of the bone (Figure 1i). Similar to *Platecarpus* and *Clidastes*, the two structures are not fully enclosed by bones and the frontal cranial crests delimit only their lateral margins (Figure 1i). The ventral margin of the olfactory bulbs and peduncles is delineated here from the dorsa-ventral depth of the frontal cranial crests. The anterior end of the olfactory bulbs is not well delimited and it cannot be distinguished from the course of the olfactory nerves. As in *Clidastes* and *Platecarpus* [58], the imprints left on the ventral surface of the frontal in *Tethysaurus* (Figure 1i) cannot help to delimit the anterior-most extent of the olfactory bulbs.

Posterior to the olfactory peduncles, the cerebral hemispheres expand laterally and ventrally to form the largest part of *Tethysaurus* endocast (Figure 1c,d). The anterior limit of the cerebral hemispheres is difficult to locate and is indicated only by changes in the width of the endocast (Figure 1c). There is no indication allowing us to delimit the cerebral hemispheres posteriorly and the whole structure is poorly differentiated from the neighboring endocast regions. As in *Clidastes* and *Platecarpus* [58], the cerebral hemispheres in *Tethysaurus* are roofed anteriorly by the posterior part of the frontal and posteriorly by the parietal (Figure 1i). The cerebral hemispheres extend between the lateral cranial crests of the parietal (Figure 1i); however, both the lateral and ventral margins of the structure are difficult to delimit due to the lack of osseous elements. Here, the ventral margin of the cerebral hemispheres in SMU 76335 (Figure 1d) was reconstructed by interpolating between the known preserved surfaces of the ventral diencephalon and the preserved olfactory peduncles and then validated against several extant taxa for plausibility.

On the dorsal surface of the *Tethysaurus* endocast (Figure 1c), a small bulge lies dorso-medial to the cerebral hemispheres and is identified as the parietal eye (*sensu* [59]) as it coincides with the median parietal foramen. The parietal eye is relatively small, as in *Clidastes* [28] (Figure 1a), but is different from the large structure reported in *Platecarpus* [28,58] (Figure 1b).

In lateral view, the part of *Tethysaurus* endocast that extends postero-ventral from the cerebral hemispheres, is identified as the ventral diencephalon (Figure 1d). It includes several structures (i.e., hypothalamus, pituitary gland, optic tracts, and optic chiasm [40]) that cannot be observed from the endocast. The whole structure lies dorsal to the rostral process of the parabasisphenoid and the posteroventral part of the ventral diencephalon is situated within the sella turcica and pituitary fossa. The ventral diencephalon in *Tethysaurus* (Figure 1d,f,h) projects more ventrally than the *medulla oblongata*, whereas the opposite condition was figured in *Platecarpus* [28,58] (Figure 1b).

Posterior to the cerebral hemispheres, the optic tectum lies on the ventral surfaces of the posterior parietal. The exact delimitation of the structure is not possible from the endocast and its position is only indicated by a marked change in width, relative to the cerebral hemispheres (Figure 1c,e,g). The optic tectum exhibits a smooth and flattened dorsal surface and lies almost on the same axis as the cerebral hemispheres (Figure 1d). Ventral to the optic tectum, the undifferentiated mesencephalon (i.e., optic tectum, torus semicircularis, and tegmentum) forms a concave margin (Figure 1d,f,h) just posterior to the ventral diencephalon.

Posterior to the mesencephalon, the rhombencephalon forms the posterior-most region of *Tethysaurus* endocast. As in *Clidastes* and *Platecarpus* [28,58], the cerebellum is not discernable from *Tethysaurus* endocast, as a cartilaginous bridge, spanning between the dorsal portions of the otic capsules, covers the structure (Figure 1d,f,h). The medulla oblongata forms a ventrally convex wide arc and exhibits a slight pontine flexure as it extends dorsally to connect with the spinal cord at the level of the foramen magnum (Figure 1d,f,h). The medulla oblongata is ventrally bordered by the posterior part of the para-basisphenoid and the basioccipital, laterally by the prootic and the opisthotic-exoccipital, and dorsally by the supraoccipital.

*3.2. Statistical Results and Morphospace Distribution*

Tests of the phylogenetic signal revealed a significant effect of evolutionary kinship on patterns of shape variation in extant toxicoferans (K-mult = 0.4012, P = 0.0001, $Z_{CR}$ = 9.4727). However, despite statistical significance, the low K statistic value indicates a weak phylogenetic structuring.

The Procrustes regression analysis showed a significant effect of allometry on endocast shape in extant toxicoferans, with allometry explaining 13.8% of shape variation (Table 2). Procrustes ANOVA (PLM) performed using allometry-corrected shape data revealed significant differences between snakes, anguimorphs, and iguanians, with the clade category explaining 27% of endocast shape variation (Table 2). Pairwise comparisons revealed that the endocast shape in each clade of extant toxicoferans differs from that of the other two clades, with the greatest distance observed between snakes and iguanians (Table 3).

Principal Component Analysis of the endocast morphology of *Tethysaurus*, anguimorphs, snakes, and iguanians resulted in the two first PCs of morphospace accounting for 51.6% of shape variance (PC1 = 35%, PC2 = 16.6%; Figure 2, see Supplementary Data S3 for the variance explained by other PCs). Along the two PCs, the position of *Tethysaurus* is distinct from the different distribution areas of the three clades of extant squamates.

The distribution along PC1 distinguishes the three extant clades—iguanians, anguimorphs, and snakes—from each other (Figure 2), though there are some exceptions. Iguanians show the greatest dispersion across morphospace and represent the extreme values along PC1. *Tethysaurus* falls within the range of values seen in some anguimorphs on the negative side of this axis. PC1 mostly captures morphological variation in the olfactory complex and cerebral hemispheres, with the negative side possessing relatively long

and slender olfactory bulbs and peduncles that project antero-ventrally. In these species, the anterior end of the olfactory bulbs is wider than the posterior part of the olfactory peduncles. In the dorsal view, the most lateral point of the cerebral hemispheres in these species is located on the postero-dorsal half of the structure. In contrast, species positioned on the positive side of PC1, mostly serpentes, show relatively short olfactory bulbs and peduncles. In these species, the widest portion of the olfactory complex is situated more posteriorly and maintains a consistent width toward the posterior end of the structure. In the dorsal view, species on the positive side of PC1 display cerebral hemispheres where the most lateral point is located on the anterior half of the structure. Additionally, the ventral diencephalon in the species on the negative side of PC1 projects more ventrally than the medulla oblongata in the lateral view. This differs from species on the positive side of PC1 in which the ventral diencephalon is aligned with the ventral margin of the medulla oblongata.

**Table 2.** Results of (1) the Procrustes regression for the test of scaling endocast shape data with size based on log-transformed centroid sizes and (2) the Procrustes ANOVA (PLM) obtained from the allometry-corrected shape data for the test of differences in endocast shape between iguanians, snakes, and anguimorphs.

| Models | Df | SS | MS | Rsq | F | Z | p |
|---|---|---|---|---|---|---|---|
| | | | Procrustes allometric regression | | | | |
| log(GPA$Csize) | 1 | 0.21119 | 0.211185 | 0.13804 | 12.011 | 4.972 | 0.0001 |
| | | | Procrustes ANOVA | | | | |
| Clades | 2 | 0.35638 | 0.178188 | 0.27024 | 13.702 | 6.3235 | 0.0001 |

Note: Significance test was based on 10,000 iterations.

**Table 3.** Pairwise comparisons of allometry-corrected shape disparity between endocasts of extant toxicoferans.

| | d | UCL (95%) | Z | Pr > d |
|---|---|---|---|---|
| Anguimorphs:Iguanians | 0.1027768 | 0.06384204 | 3.395855 | 0.0002 |
| Anguimorphs:Snakes | 0.1032785 | 0.04825282 | 4.346074 | 0.0001 |
| Iguanians:Snakes | 0.1666223 | 0.06216106 | 5.140798 | 0.0001 |

Note: Significance test was based on 10,000 iterations.

The distribution along PC2 (Figure 2) does not separate iguanians, anguimorphs, and snakes from each other. *Tethysaurus* falls within the most negative range of snakes on that axis. Species along the negative side of PC2 exhibit a relatively narrow endocast in the dorsal view that is flattened in the lateral view. In these species, the anterior end of the olfactory bulbs is distant from the most lateral point of the olfactory complex, the most lateral point of the endocast is on the anterior half of the cerebral hemispheres, and the most dorsal point of the endocast is reached at the level of the rhombencephalon. In contrast, species positioned on the positive side of PC2 exhibit wide endocasts that are dorso-ventrally taller in the lateral view. In these species, the most lateral and dorsal points of the endocast are located on the posterior half of the cerebral hemispheres. In addition, the ventral diencephalon in species on the negative side of PC2 shows no marked ventral projection, being at the same level as the mesencephalic ventral margin. This differs from species on the positive side of PC2 in which the ventral diencephalon is distinct from the mesencephalic ventral margin.

Results obtained from the Linear Discriminant Analysis (LDA) show that the first discriminant function (DF1) explains 86.54% of the total variance, whereas DF2 captures 13.46% (Figure 3). The accuracy of the LDA to classify modern species as either an anguimorph, iguanian, or snake is 93.5%, and only four species, *Lanthanotus borneensis*, *Amerotyphlops brongersmianus*, *Chamaeleo calyptratus*, and *Stenocercus roseiventris* were incorrectly assigned (see Supplementary Data S4 for details). Our results indicate that the *Tethysaurus* endocast

shape exhibits more resemblance with the endocast of snakes (0.99 posterior probability) than with anguimorhs' ($1.76 \times 10^{-4}$) and iguanians' ($1.73 \times 10^{-7}$).

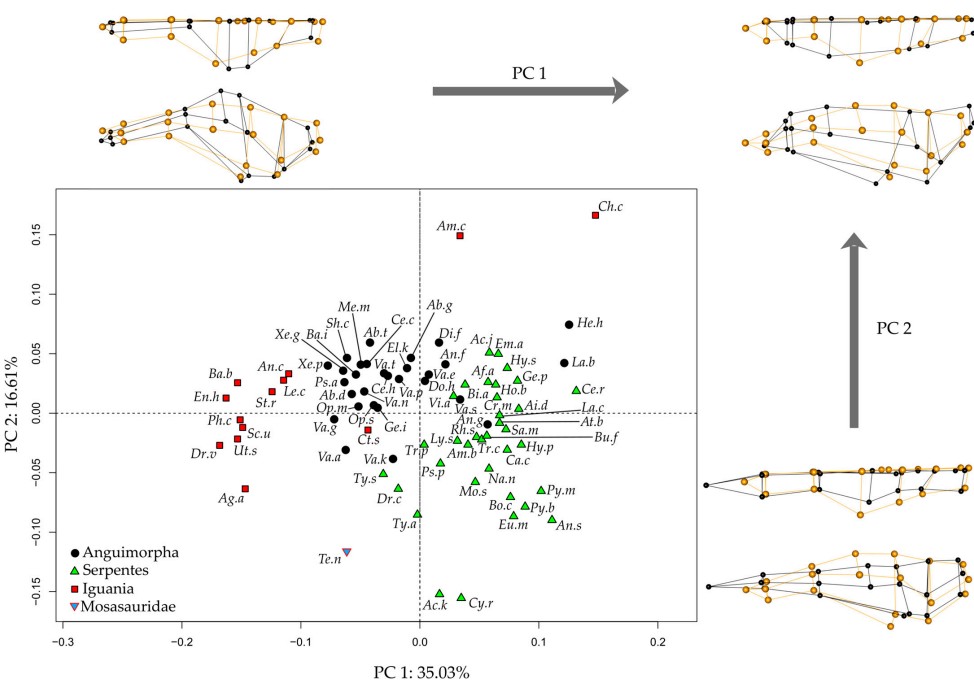

**Figure 2.** Morphospaces of endocast shape between *Tethysaurus* and extant squamates. The two first principal component axes PC1-PC2 are visualized. The wireframes represent the difference in landmark position between the most extreme endocranial morphologies along PC1 and PC2 (figured in black) as compared to the average (figured in orange), in dorsal (**top**) and lateral (**bottom**) views. Abbreviations of the species names are in Table 1.

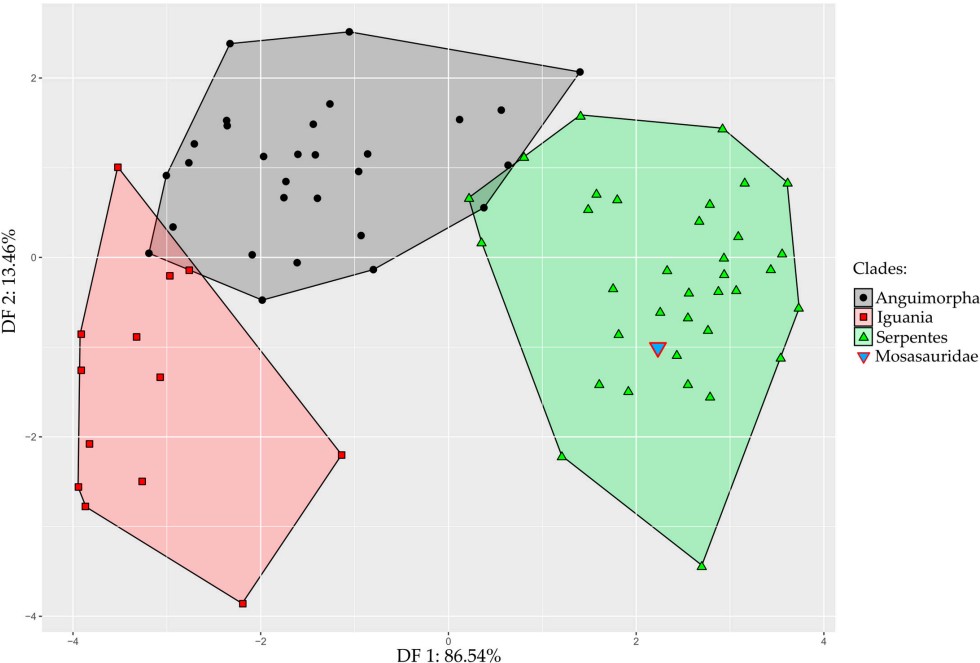

**Figure 3.** Linear Discriminant Analysis of endocranial shape variation in toxicoferans squamates, constructed from DF1 and DF2. Area of distribution of iguanians, anguimorphs, and snakes are figured in red, black, and green, respectively. The inverted blue triangle indicates the position of the mosasaurid *Tethysaurus nopcsai*.

## 4. Discussion and Conclusions

Paleoneurological studies dealing with mosasaurid endocasts remain rare. Using micro-computed tomography, for the first time, we reconstruct the brain endocast of a mosasaurid squamate, namely the early *Tethysaurus nopcsai* from the Turonian of Morocco. Our digital reconstructions showed that *Tethysaurus* is characterized by a relatively narrow and flattened endocast with weak cephalic and pontine flexures. The olfactory bulbs and peduncles are relatively long and gracile, with the anterior part of the olfactory complex being wider than the posterior part. The cerebral hemispheres represent the most dorsal and lateral points of the endocast. Posterior to the cerebral hemispheres, the position of the optic tectum is only indicated by gradual changes in the endocast width. The optic tectum lies almost on the same axis as the cerebral hemispheres. The ventral diencephalon projects more ventrally than the ventral margin of the medulla oblongata. Finally, the posterior end of the slightly convex medulla oblongata remains ventral to the antero-posterior axis the olfactory bulbs and peduncles.

Comparisons between *Tethysaurus* and the later *Platecarpus* and *Clidastes* suggest that different endocranial organizations likely occurred among mosasaurids. Both *Tethysaurus* and *Clidastes* exhibit elongated medio-laterally compressed olfactory bulbs and peduncles in the middle of their antero-posterior length, along with a relatively small parietal eye. However, as Camp [28] only figured out the dorsal view of *Clidastes* endocast, comparisons are limited and restricted to these details. In contrast, the endocranial organization in *Tethysaurus* seems different from that reported in *Platecarpus*. The latter, as figured by Camp [28], exhibits short olfactory bulbs and peduncles, a relatively large parietal eye, and a pronounced cephalic flexure differing from *Tethysaurus* and *Clidastes*. It is worth noting that the olfactory bulbs and peduncles in *Platecarpus* were figured in dotted lines by Camp [28] and Russell [58], making the exact length of the whole structure uncertain. Thus, the most notable distinctions between *Tethysaurus* and *Platecarpus* consist of the endocast flexure and the size of the parietal eye.

Pronounced brain flexures generally characterize the early ontogenetic stages of brain development in non-avian reptiles (e.g., [60–62]). When the cranial cavity is relatively small, the brain shows a more pronounced 'S' shape, with the anterior part positioned higher than the posterior one [34]. As the braincase grows more rapidly than the brain itself, there is more space available, allowing the brain to 'unfold' in mature individuals [63–65]. Here, the variable degrees of flexure noticed between *Tethysaurus* and *Platecarpus* might be potentially related to different ontogenetic stages. The endocast flexures observed in *Platecarpus* specimen UCMP 34781 are consistent with the late juvenile stage reported by Sheldon [66]. In contrast, although the ontogenetic stage for *Tethysaurus* remains undetermined [48], the specimen is presumably an adult on the basis of several anatomical characteristics [67] and the nearly straight endocast reported here reinforces such a hypothesis. However, it is worth noting that, as in extant archosaurs [61], possible heterochrony may impact the morphology of the mosasaurid endocast. Thus, the pronounced flexure observed in *Platecarpus* could possibly indicate retention of juvenile endocranial shape (i.e., paedomorphosis) in an adult specimen. In this context, further studies investigating changes in mosasaurid endocast morphology over ontogeny are required. Although ontogenetic series of mosasaurid skulls are rather scarce, future studies could consider the limited material already available (e.g., *Tylosaurus*, [68]) to assess the extent to which endocast morphology in mosasaurids reflects ontogenetic stages.

The variable position and size of the parietal eye have been reported within Mosasauridae (e.g., [69–73]), including differences at the intraspecific level (e.g., *Tylosaurus proriger*, [74]). Such variability in the size of the parietal eye was also reported in various clades of vertebrates (e.g., [75]), suggesting that the common occurrence of a large or small foramen is not a phylogenetic signal but is instead related to differences in the photoreceptive and neuroendocrine functions associated with the structure [76]. The parietal eye in extant squamates detects changes in light levels and this sensory input regulates various seasonal metabolic processes, including behavioral thermoregulation, diurnal rhythms, physical

activity, and behavior (e.g., [77,78]). In mosasaurids, the environmental conditions and physiological roles associated with the development of a large parietal eye remain unclear. Connolly [73] found no significant relationships between the size of the parietal eye and the paleolatitude distribution of mosasaurids, nor between the size of the parietal eye and their ability to dive deep. In this context, Connolly [73] suggested that the mosasaurid parietal eye may have functioned primarily for navigation and orientation related to migration. However, such correlations remain unclear and physiological roles associated with the variability in the reptilian parietal eye are needed to better interpret the variability in mosasaurids.

The results obtained here showed that the endocast morphology in extant toxicoferans is linked to phylogeny. However, despite being statistically significant, the low K statistic value indicates weak phylogenetic structuring. Similar to previous studies conducted on snake endocasts [29,33], this indicates that, although significant, the phylogenetic signal remains weak and other factors (e.g., habitat, activity period) could influence the endocast morphology in all extant toxicoferans.

Comparisons with extant toxicoferans highlighted the unique combination of endocranial features in *Tethysaurus*, showing only partial resemblance with anguimorphs, iguanians, or snakes. Indeed, the endocast of *Tethysaurus* does share morphological similarities with each of these three clades, such as (1) the relatively long and gracile olfactory bulbs and peduncles in *Tethysaurus* resemble those observed in iguanians and some anguimorphs, contrasting with the wider and shorter structures seen in snakes; (2) the weak cephalic flexure observed in *Tethysaurus* is similar to that found in anguimorphs and snakes, whereas most iguanians typically exhibit a stronger flexure in their endocast; and (3) posterior to the cerebral hemispheres, the gradual changes in endocast width indicating the position of the optic tectum in *Tethysaurus* resembles the condition observed in anguimorphs but differs from the more abrupt narrowing seen in snakes or the nearly absence of change in iguanians. Overall, the strong endocranial resemblance between varanids and mosasaurids reported by Camp [28] is not observed here. Instead, our results suggested more similarities in endocast shape between *Tethysaurus* and some of the snakes. To assess this result, further comparisons should expand data sampling on mosasaurid endocasts to provide a more comprehensive understanding of the morphological variability within the clade. Thus, the CT data already existing for the mosasaurid *Plotosaurus bennisoni*, available from the online database MorphoSource, could constitute a good starting point for such endocranial studies.

The position of mosasaurids within Toxicofera varies depending on phylogenetic analysis. The clade is positioned either within Anguimorpha and closely related to varanoids (e.g., [45,47]), as the sister group of snakes (e.g., [42,79]), or as the sister group of a clade comprising Anguimorpha and Iguania (e.g., [43,80]). Camp [28] suggested a close phylogenetic relationship between mosasaurids and varanids as he observed a strong resemblance in the morphology of their endocasts. However, this hypothesis is challenged here, as we observe similarities in endocast shape between *Tethysaurus* and each of the three clades included in Toxicofera, with a particular resemblance to some snakes (*Acutotyphlops kunuaensis*, *Typhlops arenarius*, *Anilius scytale*, *Cylindrophis ruffus*, *Eunectes murinus,* and *Python bivittatus*). These results, based on the digital brain endocast alone, do not enable this study to position mosasaurids within Toxicofera and, furthermore, our findings support no specific phylogenetic hypothesis. In this context, phylogenetic inferences made from endocast morphologies should be treated with caution as other factors may influence the endocast morphology in squamates. In addition, accurate and precise inferences of brain morphology from mosasaurid endocasts require careful consideration. Although certain aspects of brain morphologies in squamates can be extrapolated from endocast morphology, such information varies according to species, clades, and brain regions [40]. Therefore, predicting brain–endocast ratios in mosasaurids, identifying which parts of the endocast accurately reflect brain morphology, and enabling biological inferences require the consideration of a large panel of extant toxicoferans in order to avoid any misinterpretations.

**Supplementary Materials:** The following supporting information can be downloaded at https://www.mdpi.com/article/10.3390/d16090548/s1. Figure S1: Schematic phylogenetic relationships of iguanians sampled in the study (modified from [50,52]) associated with 3D renderings showing the endocast in dorsal (left) and left lateral (right) views. Figure S2: Schematic phylogenetic relationships of anguimorphs sampled in the study (modified from [50,52]) associated with 3D renderings showing the endocast in dorsal (left) and left lateral (right) views. Figure S3: Schematic phylogenetic relationships of snakes sampled in the study (modified from [50,52]) associated with 3D renderings showing the endocast in dorsal (left) and left lateral (right) views. Supplementary Data S1: Landmark description used in this study (modified from [30]) and position on *Tethysaurus* endocast in dorsal and lateral views. Supplementary Data S2: Raw coordinates of the 19 landmarks placed on each species used in the study. Abbreviations in the main text (Table 1). Supplementary Data S3: Variance explained along the different PCs obtained in the Principal Component Analysis performed on the endocasts of *Tethysaurus* and extant squamates. Supplementary Data S4: Results LDA.

**Author Contributions:** Conceptualization, R.A. and N.B.; methodology, R.A. and C.L.-A.; software, R.A.; investigation, R.A.; writing—original draft preparation, R.A.; writing—review and editing, M.J.P., A.H., P.V., C.L.-A., and N.B.; visualization, R.A.; supervision, A.H., P.V., and N.B.; funding acquisition, N.B., A.H., and P.V. All authors have read and agreed to the published version of the manuscript.

**Funding:** This work was supported by a grant from the Agence Nationale de la Recherche under the LabEx ANR-10-LABX-0003-BCDiv, in the program 'Investissements d'avenir' ANR-11-IDEX-0004-02.

**Institutional Review Board Statement:** Not applicable.

**Data Availability Statement:** The endocasts (stl files) used in the study will be available on MorphoSource and attached to the corresponding files for each species. The raw scan data of *Tethysaurus* specimen MNHN GOU 1 will be available on the MNHN Digitization Work Portal https://3dtheque.mnhn.fr (accessed on 4 August 2024).

**Acknowledgments:** We thank the AST-RX, plateau d'Accès Scientifique à la Tomographie à Rayons X du MNHN, UAR 2700 2AD CNRS-MNHN, Paris, for access to the CT scanner and M. Garcia-Sanz (MNHN, UMS 2700 OMSI) for producing the CT scan of specimen MNHN GOU 1. We also thank M. Colbert and the High-Resolution X-ray Computed Tomography Facility at The University of Texas for producing CT scans of the specimens SMU 76335 and SMU 75486. We warmly thank I. Ineich, J. Rosado, K. Lim, and A. Herrel for the loan of CT scans. We would like to acknowledge L.A. Scheinberg, S. Scarpetta, C. Spencer, D. Ledesma, E. Stanley, A. Motta, G. Schneider, N. Rios, S. Grant, J. Maisano, Z. Randall, J. Gray, C. Sheehy, and D. Blackburn for access to CT scans through the Morphosource database. We would also like to thank two anonymous reviewers for their helpful comments and suggestions that improved the paper.

**Conflicts of Interest:** The authors declare no conflicts of interest.

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
