# Peer review of "First Virtual Reconstruction of a Mosasaurid Brain Endocast: Description and Comparison of the Endocast of Tethysaurus nopcsai with Those of Extant Squamates"

_diversity, doi:10.3390/d16090548_

Round 1

Reviewer 1 Report

Comments and Suggestions for Authors

To the editors and the author

- I am very happy to see that the reconstruction material of a mosasaurid brain endocast is described. This topic is still very little developed and it is extremely important to carry out this type of study. I recommend the publication in diversity.

I did not mark the review PDF, but I will summarize my comments below, mainly about adding a bibliography in the introduction.

Line 53: please add

Fernández Marta S. & Talevi Marianella. 2015. An halisaurine (Squamata: Mosasauridae) from the Late Cretaceous of Patagonia, with a preserved tympanic disc: insights into mosasaur middle ear. Comptes rendus Palevol14: 483–493.

Line 56:

please add

P. González Ruiz, M. Férnandez, M. Talevi, J. Leardi, M. Reguero.2019. A new Plotosaurini mosasaur skull from the upper Maastrichtian of Antarctica. Plotosaurini paleogeographic occurrences. Cretaceous Research 100, 104166-10417. https://doi.org/10.1016/j.cretres.2019.06.012

Author Response

Dear Reviewer 1:

- I did not mark the review PDF, but I will summarize my comments below, mainly about adding a bibliography in the introduction.

Line 53: please add Fernández Marta S. & Talevi Marianella. 2015. An halisaurine (Squamata: Mosasauridae) from the Late Cretaceous of Patagonia, with a preserved tympanic disc: insights into mosasaur middle ear. Comptes rendus Palevol14: 483–493.

Line 56: please add González Ruiz, M. Férnandez, M. Talevi, J. Leardi, M. Reguero.2019. A new Plotosaurini mosasaur skull from the upper Maastrichtian of Antarctica. Plotosaurini paleogeographic occurrences. Cretaceous Research 100, 104166-10417. https://doi.org/10.1016/j.cretres.2019.06.012

The two references were added.

Thank you

Best regards,

Rémi Allemand

Reviewer 2 Report

Comments and Suggestions for Authors

Comments specific to particular lines of text or figures have been made on a file of the manuscript.

More general comments are as follows:

This study provides the first interpretation of a mosasaur endocast based on CT scans.  The endocast of Tethysaurus is compared to other toxicoferan taxa within Anguimorpha, Iguanidae, and Serpentes, and to previously made latex casts of two mosasaurs.  Based on the analyses, close similarities were not found between Tethysaurus and varanids, and some snakes were found to be the most similar to Tethysaurus.  Overall, this is an interesting study, and, from my perspective, the methodology seems appropriate.  The authors could consider including a brief discussion about why including other taxa for which CT scans already exist (such as Plotosaurus) was beyond the scope of this initial investigation.  

Regarding the comment about additional references in the middle of the first paragraph (comment associated with line 51), naturally, I do not expect you to cite a reference about every mosasaur ever described, but I think either a selection of broad-scope / review-style papers or a selection of papers describing mosasaurs with some unique feeding or locomotor adaptations would be beneficial.  Some examples of the latter category (unfortunately also all biased toward later Campanian – Maastrichtian taxa) include:  Any of Lindgren's work on Plotosaurus (eg. Lindgren et al. 2007 or 2008), any of Konishi's work on Japanese taxa (eg. Konishi et al. 2015 or 2023), or Strong et al. 2020.  By making this recommendation, I am not expecting these exact references to be cited, they are merely examples.

Regarding the supplementary information files:

Consider adding simple labels to supplementary figures 1-3 at the tops of the columns of endocast reconstructions saying something along the lines of "Dorsal view" and "Left-lateral view".

Additionally, I am aware that the complete set of landmarks used in this study is provided in a fully illustrated and annotated format in Allemand et al. 2023a, but perhaps consider including a simplified list as an additional supplementary file for readers of this article.  

Comments on the Quality of English Language

Overall, the quality of the English language is excellent.  There are a few minor errors here and there.  These mostly involve preposition choice, which I acknowledge can be very difficult.  

Author Response

Please see the word document attached.
